# Adversarial Ranking for Language Generation

**Kevin Lin**[*]
University of Washington
kvlin@uw.edu

**Dianqi Li**[*]
University of Washington
dianqili@uw.edu

**Xiaodong He**
Microsoft Research
xiaohe@microsoft.com

**Zhengyou Zhang**
Microsoft Research
zhang@microsoft.com

**Ming-Ting Sun**
University of Washington
mts@uw.edu

## Abstract

Generative adversarial networks (GANs) have great successes on synthesizing data. However, the existing GANs restrict the discriminator to be a binary classifier, and thus limit their learning capacity for tasks that need to synthesize output with rich structures such as natural language descriptions. In this paper, we propose a novel generative adversarial network, RankGAN, for generating high-quality language descriptions. Rather than training the discriminator to learn and assign absolute binary predicate for individual data sample, the proposed RankGAN is able to analyze and rank a collection of human-written and machine-written sentences by giving a reference group. By viewing a set of data samples collectively and evaluating their quality through relative ranking scores, the discriminator is able to make better assessment which in turn helps to learn a better generator. The proposed RankGAN is optimized through the policy gradient technique. Experimental results on multiple public datasets clearly demonstrate the effectiveness of the proposed approach.

## 1 Introduction

Language generation plays an important role in natural language processing, which is essential to many applications such as machine translation [1], image captioning [6], and dialogue systems [26]. Recent studies [10, 11, 29, 33] show that the recurrent neural networks (RNNs) and the long short-term memory networks (LSTMs) can achieve impressive performances for the task of language generation. Evaluation metrics such as BLEU [22], METEOR [2], and CIDEr [32] are reported in the literature.

Generative adversarial networks (GANs) have drawn great attentions since Goodfellow *et al.* [8] introduced the framework for generating the synthetic data that is similar to the real one. The main idea behind GANs is to have two neural network models, the discriminator and the generator, competing against each other during learning. The discriminator aims to distinguish the synthetic from the real data, while the generator is trained to confuse the discriminator by generating high quality synthetic data. During learning, the gradient of the training loss from the discriminator is then used as the guidance for updating the parameters of the generator. Since then, GANs achieve great performance in computer vision tasks such as image synthesis [5, 14, 17, 24, 27]. Their successes are mainly attributed to training the discriminator to estimate the statistical properties of the continuous real-valued data (e.g., pixel values).

---

[*]The authors contributed equally to this work.

The adversarial learning framework provides a possible way to synthesize language descriptions in high quality. However, GANs have limited progress with natural language processing. Primarily, the GANs have difficulties in dealing with discrete data (e.g., text sequences [3]). In natural languages processing, the text sequences are evaluated as the discrete tokens whose values are non-differentiable. Therefore, the optimization of GANs is challenging. Secondly, most of the existing GANs assume the output of the discriminator to be a binary predicate indicating whether the given sentence is written by human or machine [4, 16, 18, 34, 35]. For a large variety of natural language expressions, this binary predication is too restrictive, since the diversity and richness inside the sentences are constrained by the degenerated distribution due to binary classification.

In this paper, we propose a novel adversarial learning framework, RankGAN, for generating high-quality language descriptions. RankGAN learns the model from the relative ranking information between the machine-written and the human-written sentences in an adversarial framework. In the proposed RankGAN, we relax the training of the discriminator to a learning-to-rank optimization problem. Specifically, the proposed new adversarial network consists of two neural network models, a generator and a ranker. As opposed to performing a binary classification task, we propose to train the ranker to rank the machine-written sentences lower than human-written sentences with respect to a reference sentence which is human-written. Accordingly, we train the generator to synthesize sentences which confuse the ranker so that machine-written sentences are ranked higher than human-written sentences in regard to the reference. During learning, we adopt the policy gradient technique [31] to overcome the non-differentiable problem. Consequently, by viewing a set of data samples collectively and evaluating their quality through relative ranking, the discriminator is able to make better assessment of the quality of the samples, which in turn helps the generator to learn better. Our method is suitable for language learning in comparison to conventional GANs. Experimental results clearly demonstrate that our proposed method outperforms the state-of-the-art methods.

## 2   Related works

**GANs:** Recently, GANs [8] have been widely explored due to its nature of unsupervised deep learning. Though GANs achieve great successes on computer vision applications [5, 14, 17, 24, 27], there are only a few progresses in natural language processing because the discrete sequences are not differentiable. To tackle the non-differentiable problem, SeqGAN [35] addresses this issue by the policy gradient inspired from the reinforcement learning [31]. The approach considers each word selection in the sentence as an action, and computes the reward of the sequence with the Monte Carlo (MC) search. Their method back-propagates the reward from the discriminator, and encourages the generator to create human-like language sentences. Li *et al.* [18] apply GANs with the policy gradient method to dialogue generation. They train a Seq2Seq model as the generator, and build the discriminator using a hierarchical encoder followed by a 2-way softmax function. Dai *et al.* [4] show that it is possible to enhance the diversity of the generated image captions with conditional GANs. Yang *et al.* [34] further prove that training a convolutional neural network (CNN) as a discriminator yields better performance than that of the recurrent neural network (RNN) for the task of machine translation (MT). Among the works mentioned above, SeqGAN [35] is the most relevant study to our proposed method. The major difference between SeqGAN [35] and our proposed model is that we replace the regression based discriminator with a novel ranker, and we formulate a new learning objective function in the adversarial learning framework. In this condition, the rewards for training our model are not limited to binary regression, but encoded with relative ranking information.

**Learning to rank:** Learning to rank plays an essential role in Information Retrieval (IR) [21]. The ranking technique has been proven effective for searching documents [12] and images [23]. Given a reference, the desired information (such as click-through logs [15]) is incorporated into the ranking function which aims to encourage the relevant documents to be returned as early as possible. While the goal of previous works is to retrieve relevant documents, our proposed model takes the ranking scores as the rewards to learn the language generator. Our proposed RankGAN is one of the first generative adversarial network which learns by relative ranking information.

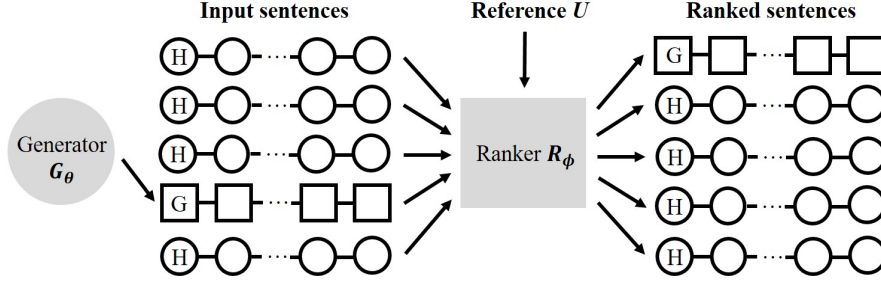

Figure 1: An illustration of the proposed RankGAN. **H** denotes the sentence sampled from the human-written sentences. **G** is the sentence generated by the generator $G_\theta$. The inputs of the ranker $R_\phi$ consist of one synthetic sequence and multiple human-written sentences. Given the reference sentence $U$ which is written by human, we rank the input sentences according to the relative scores. In this figure, it is illustrated that the generator tries to fool the ranker and let the synthetic sentence to be ranked at the top with respect to the reference sentence.

## 3 Method

### 3.1 Overall architecture

In conventional GANs [8], the discriminator with multilayer perceptrons outputs a binary probability distribution to suggest whether the unknown sequences come from the real data rather than the data synthesized by a generator. In contrast to conventional GANs, RankGAN consists of a sequence generator $G$ and a ranker $R$, where the $R$ can endow a relative rank among the sequences when given a reference. As illustrated in Figure 1, the learning objective of $G$ is to produce a synthetic sentence that receives higher ranking score than those drawn from real data. However, the goal of $R$ is to rank the synthetic sentence lower than human-written sentences. Thus, this can be treated as $G$ and $R$ play a minimax game with the objective function $\mathfrak{L}$:

$$\min_{\theta} \max_{\phi} \mathfrak{L}(G_\theta, R_\phi) = \mathbb{E}_{s \sim \mathcal{P}_h} \left[ \log R_\phi(s|U, \mathcal{C}^-) \right] + \mathbb{E}_{s \sim G_\theta} \left[ \log(1 - R_\phi(s|U, \mathcal{C}^+)) \right] \quad (1)$$

where $\theta$ and $\phi$ are the variable parameters in $G$ and $R$, respectively. The $\mathbb{E}$ is the expectation operator, and $\mathcal{P}_h$ is the real data from human-written sentences. $s \sim \mathcal{P}_h$ and $s \sim G_\theta$ denote that $s$ is from human-written sentences and synthesized sentences, respectively. The $U$ is the reference set used for estimating relative ranks, and $\mathcal{C}^+, \mathcal{C}^-$ are the comparison set with regard to different input sentences $s$. When the input sentence $s$ is the real data, $\mathcal{C}^-$ is pre-sampled from generated data; If the input sentence $s$ is the synthetic data, the $\mathcal{C}^+$ is pre-sampled from human-written data.

The forms of $G_\theta$ and $R_\phi$ can be achieved in many ways. In this paper, we design the generative model with the long short-term memory networks (LSTMs) [11]. A LSTM iteratively takes the embedded features of the current token $w_t$ plus the information in the hidden state $h_{t-1}$ and the cell state $c_{t-1}$ from previous stages, and updates the current states $h_t$ and $c_t$. Additionally, the subsequent word $w_{t+1}$ is conditionally sampled subjects to the probability distribution $p(w_{t+1}|h_t)$ which is determined by the value of the current hidden state $h_t$. Benefiting from the capacity of LSTMs, our generative model can conserve long-term gradient information and produce more delicate word sequences $s = (w_0, w_1, w_2, ..., w_T)$, where $T$ is the sequence length.

Recent studies show that the convolutional neural network can achieve high performance for machine translation [7, 34] and text classification [36]. The proposed ranker $R$, which shares the similar convolutional architecture, first maps concatenated sequence matrices into the embedded feature vectors $y_s = \mathfrak{F}(s)$ through a series of nonlinear functions $\mathfrak{F}$. Then, the ranking score will be calculated for the sequence features $y_s$ with the reference feature $y_u$ which is extracted by $R$ in advance.

## 3.2 Rank score

More disparities between sentences can be observed by contrasts. Inspired by this, unlike the conventional GANs, our architecture possesses a novel comparison system that evaluates the relative ranking scores among sentences. Following the ranking steps commonly used in Web search [12], the relevance score of the input sequence $s$ given a reference $u$ is measured as:

$$\alpha(s|u) = cosine(y_s, y_u) = \frac{y_s \cdot y_u}{\|y_s\|\|y_u\|} \tag{2}$$

where the $y_u$ and $y_s$ are the embedded feature vectors of the reference and the input sequence, respectively. $\|\cdot\|$ denotes the norm operator. Then, a softmax-like formula is used to compute the ranking score for a certain sequence $s$ given a comparison set $\mathcal{C}$:

$$P(s|u, \mathcal{C}) = \frac{exp(\gamma\alpha(s|u))}{\sum_{s' \in \mathcal{C}'} exp(\gamma\alpha(s'|u))} \tag{3}$$

The parameter $\gamma$, whose value is set empirically during experiments, shares the similar idea with the Boltzmann exploration [30] method in reinforcement learning. Lower $\gamma$ results in all sentences to be nearly equiprobable, while higher $\gamma$ increases the biases toward the sentence with the greater score. The set $\mathcal{C}' = \mathcal{C} \cup \{s\}$ denotes the set of input sentences to be ranked.

The collective ranking score for an input sentence is an expectation of its scores given different references sampled across the reference space. During learning, we randomly sample a set of references from human-written sentences to construct the reference set $U$. Meanwhile, the comparison set $\mathcal{C}$ will be constructed according to the type of the input sentence $s$, i.e., $\mathcal{C}$ is sampled from the human-written set if $s$ is a synthetic sentence produced by $G$, and vice versa. With the above setting, the expected log ranking score computed for the input sentence $s$ can be derived by:

$$\log R_\phi(s|U, \mathcal{C}) = \mathbb{E}_{u \in U} \log\left[P(s|u, \mathcal{C})\right] \tag{4}$$

Here, $s$ is the input sentence. It is either human-written or produced by $G_\theta$. Accordingly, the comparison set $\mathcal{C}$ is $\mathcal{C}^+$ if $s$ is generated by machine, and vice versa. Given the reference set and the comparison set, we are able to compute the rank scores indicating the relative ranks for the complete sentences. The ranking scores will be used for the objective functions of generator $G_\theta$ and ranker $R_\phi$.

## 3.3 Training

In conventional settings, GANs are designed for generating real-valued image data and thus the generator $G_\theta$ consists of a series of differential functions with continuous parameters guided by the objective function from discriminator $D_\phi$ [8]. Unfortunately, the synthetic data in the text generation task is based on discrete symbols, which are hard to update through common back-propagation. To solve this issue, we adopt the Policy Gradient method [31], which has been widely used in reinforcement learning.

Suppose the vocabulary set is $V$, at time step $t$, the previous tokens generated in the sequence are $(w_0, w_1, ..., w_{t-1})$, where all tokens $w_i \in V$. When compared to the typical reinforcement learning algorithms, the existing sequence $s_{1:t-1} = (w_0, w_1, ..., w_{t-1})$ is the current state, the next token $w_t$ that selected in the next step is an action sampling from the policy $\pi_\theta(w_t|s_{1:t-1})$. Since we use $G$ to generate the next token, the policy $\pi_\theta$ equals to $p(w_t|s_{1:t-1})$ which mentioned previously, and $\theta$ is the parameter set in generator $G$. Once the generator reaches the end of one sequence (i.e., $s = s_{1:T}$), it receives a ranking reward $R(s|U, \mathcal{C})$ according to the comparison set $\mathcal{C}$ and its related reference set $U$.

Note that in reinforcement learning, the current reward is compromised by the rewards from intermediate states and future states. However, in text generation, the generator $G_\theta$ obtains the reward if and only if one sequence has been completely generated, which means no intermediate reward is gained before the sequence hits the end symbol. To relieve this problem, we utilize the Monte Carlo rollouts

methods [4, 35] to simulate intermediate rewards when a sequence is incomplete. Then, the expected future reward $V$ for partial sequences can be computed by:

$$V_{\theta,\phi}(s_{1:t-1}, U) = \underset{s_r \sim G_\theta}{\mathbb{E}} \left[ \log R_\phi(s_r|U, \mathcal{C}^+, s_{1:t-1}) \right] \qquad (5)$$

Here, $s_r$ represents the complete sentence sampled by rollout methods with the given starter sequence $s_{1:t-1}$. To be more specific, the beginning tokens $(w_0, w_1, ..., w_{t-1})$ are fixed and the rest tokens are consecutively sampled by $G_\theta$ until the last token $w_T$ is generated. We denote this as the "path" generated by the current policy. We keep sampling $n$ different paths with the corresponding ranking scores. Then, the average ranking score will be used to approximate the expected future reward for the current partial sequence.

With the feasible intermediate rewards, we can finalize the objective function for complete sentences. Refer to the proof in [31], the gradient of the objective function for generator $G$ can be formulated as:

$$\nabla_\theta \mathfrak{L}_\theta(s_0) = \underset{s_{1:T} \sim G_\theta}{\mathbb{E}} \left[ \sum_{t=1}^{T} \sum_{w_t \in V} \nabla_\theta \pi_\theta(w_t|s_{1:t-1}) V_{\theta,\phi}(s_{1:t}, U) \right] \qquad (6)$$

where $\nabla_\theta$ is the partial differential operator. The start state $s_0$ is the first generated token $w_0$. $\mathbb{E}_{s_{1:T} \sim G_\theta}$ is the mean over all sampled complete sentences based on current generator's parameter $\theta$ within one minibatch. Note that we only compute the partial derivatives for $\theta$, as the $R_\phi$ is fixed during the training of generator. Importantly, different from the policy gradients methods in other works [4, 20, 35], our method replaces the simple binary outputs with a ranking system based on multiple sentences, which can better reflect the quality of the imitate sentences and facilitate effective training of the generator $G$.

To train the ranker's parameter set $\phi$, we can fix the parameters in $\theta$ and maximize the objective equation (1). In practice, however, it has been found that the network model learns better by minimizing $\log(R_\phi(s|U, \mathcal{C}^+))$ instead of maximizing $\log(1 - R_\phi(s|U, \mathcal{C}^+))$, where $s \sim G_\theta$. This is similar to the finding in [25]. Hence, during the training of $R_\phi$, we maximize the following ranking objective function:

$$\mathfrak{L}_\phi = \underset{s \sim \mathcal{P}_h}{\mathbb{E}} \left[ \log R_\phi(s|U, \mathcal{C}^-) \right] - \underset{s \sim G_\theta}{\mathbb{E}} \left[ \log R_\phi(s|U, \mathcal{C}^+) \right] \qquad (7)$$

It is worthwhile to note that when the evaluating data comes from the human-written sentences, the comparison set $\mathcal{C}^-$ is sampled from the generated sentences through $G_\theta$; In contrast, if the estimating data belongs to the synthetic sentences, $\mathcal{C}^+$ consists of human-written sentences. We found empirically that this gives more stable training.

### 3.4 Discussion

Note that the proposed RankGAN has a Nash Equilibrium when the generator $G_\theta$ simulates the human-written sentences distribution $P_h$, and the ranker $R_\phi$ cannot correctly estimate rank between the synthetic sentences and the human-written sentences. However, as also discussed in the literature [8, 9], it is still an open problem how a non-Bernoulli GAN converges to such an equilibrium. In a sense, replacing the absolute binary predicates with the ranking scores based on multiple sentences can relieve the gradient vanishing problem and benefit the training process. In the following experiment section, we observe that the training converges on four different datasets, and leads to a better performance compared to previous state-of-the-arts.

## 4 Experimental results

Following the evaluation protocol in [35], we first carry out experiments on the data and simulator proposed in [35]. Then, we compare the performance of RankGAN with other state-of-the-art methods on multiple public language datasets including Chinese poems [37], COCO captions [19], and Shakespear's plays [28].

Table 1: The performance comparison of different methods on the synthetic data [35] in terms of the negative log-likelihood (NLL) scores.

| Method | MLE | PG-BLEU | SeqGAN | RankGAN |
|--------|-----|---------|--------|---------|
| NLL | 9.038 | 8.946 | 8.736 | 8.247 |

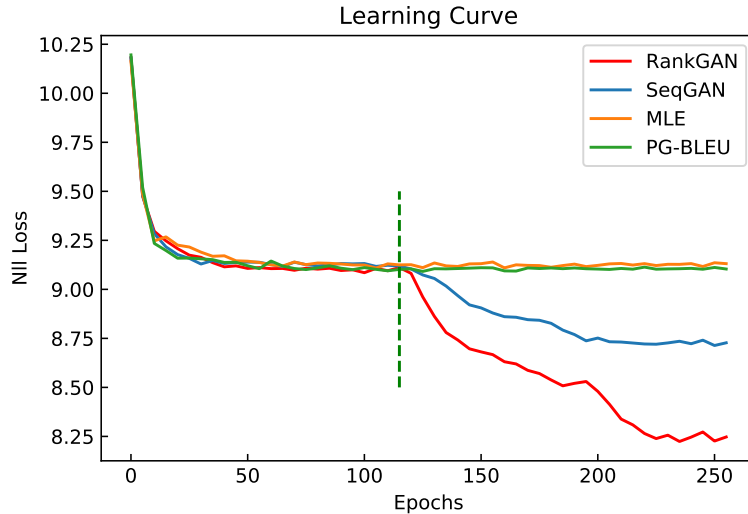

Figure 2: Learning curves of different methods on the simulation of synthetic data with respect to different training epochs. Note that the vertical dashed line indicates the end of the pre-training of PG-BLEU, SeqGAN and RankGAN.

## 4.1 Simulation on synthetic data

We first conduct the test on the dataset proposed in [35]. The synthetic data[2] is a set of sequential tokens which can be seen as the simulated data comparing to the real-word language data. We conduct this simulation to validate that the proposed method is able to capture the dependency of the sequential tokens. In the simulation, we firstly collect $10,000$ sequential data generated by the oracle model (or true model) as the training set. Note that the oracle model we used is a random initialized LSTM which is publicly available[2]. During learning, we randomly select one training sentence and one generated sentence from RankGAN to form the input set $\mathcal{C}'$. Then, given a reference sample which is also randomly selected from the training set, we compute the ranking score and optimize the proposed objective function. Note that the sentence length of the training data is fixed to 20 for simplicity.

Following the evaluation protocol in [35], we evaluate the machine-written sentences by stimulating the Turing test. In the synthetic data experiment, the oracle model, which plays the role as the human, generates the "human-written" sentences following its intrinsic data distribution $\mathcal{P}_o$. We assume these sentences are the ground truth sentences used for training, thus each model should learn and imitate the sentences from $\mathcal{P}_o$. At the test stage, obviously, the generated sentences from each model will be evaluated by the original oracle model. Following this, we take the sentences generated by RankGAN as the input of the oracle model, and estimate the average negative log-likelihood (NLL) [13]. The lower the NLL score is, the higher probability the generated sentence will pass the Turing test.

We compare our approach with the state-of-the-art methods including maximum likelihood estimation (MLE), policy gradient with BLEU (PG-BLEU), and SeqGAN [35]. The PG-BLEU computes the BLEU score to measure the similarity between the generated sentence and the human-written

Table 2: The performance comparison of different methods on the Chinese poem generation in terms of the BLEU scores and human evaluation scores.

| Method | BLEU-2 | Method | Human score |
|---|---|---|---|
| MLE | 0.667 | SeqGAN | 3.58 |
| SeqGAN | 0.738 | RankGAN | 4.52 |
| RankGAN | **0.812** | Human-written | **6.69** |

sentences, then takes the BLEU score as the reward to update the generator with policy gradient. Because PG-BLEU also learns the similarity information during training, it can be seen as a baseline comparing to our approach. It's noteworthy that while the PG-BLEU grasps the similarities depend on the n-grams matching from the token-level among sentences, RankGAN explores the ranking connections inside the embedded features of sentences. These two methods are fundamentally different. Table 1 shows the performance comparison of RankGAN and the other methods. It can be seen that the proposed RankGAN performs more favourably against the compared methods. Figure 2 shows the learning curves of different approaches with respect to different training epochs. While MLE, PG-BLEU and SeqGAN tend to converge after 200 training epochs, the proposed RankGAN consistently improves the language generator and achieves relatively lower NLL score. The results suggest that the proposed ranking objective, which relaxes the binary restriction of the discriminator, is able to learn effective language generator. It is worth noting that the proposed RankGAN achieves better performance than that of PG-BLEU. This indicates employing the ranking information as the reward is more informative than making use of the BLEU score that stands on token-level similarities. In our experiments, we noticed that the results are not sensitive to the size of comparison set and reference set. The learning curves converge to similar results with different reference sizes and comparison sizes. However, learning with the large reference size and comparison set could potentially increase the computational cost.

Conventional GANs employ a binary classifier to distinguish the human-written and the machine-created sentences. Though effective, it is also very restrictive for tasks like natural language generation, where rich structures and various language expressions need to be considered. For these tasks, usually a relative quality assessment is more suitable. The proposed RankGAN is able to perform quality assessment in a relative space, and therefore, rather than training the discriminator to assign the absolute 0 or 1 binary predicate to the synthesized or real data sample, we expect the discriminator to rank the synthetic data compared to the real data in the relative assessment space where better quality judgments of different data samples can be obtained. Given the rewards with the relative ranking information, the proposed RankGAN is possible to learn better language generator than the compared state-of-the-art methods.

## 4.2 Results on Chinese poems composition

To evaluate the performance of our language generator, we compare our method with other approaches including MLE and SeqGAN [35] on the real-word language data. We conduct experiments on the Chinese poem dataset [37], which contains $13,123$ five-word quatrain poems. Each poem has 4 sentences, and each sentence contains 5 words resulting in a total of 20 words. After the standard pre-processing which replaces the non-frequently used words (appeared less than 5 times) with the special character UNK, we train our model on the dataset and generate the poem. To keep the proposed method general, our model does not take advantage of any prior knowledge such as phonology during learning.

Following the evaluation protocol in [35, 37], we compute the BLEU-2 score and estimate the similarity between the human-written poem and the machine-created one. Table 2 summarizes the BLEU-2 score of different methods. It can be seen that the proposed RankGAN performs more favourably compared to the state-of-the-art methods in terms of BLEU-2 score. This indicates that the proposed objective is able to learn effective language generator with real-world data.

We further conduct human study to evaluate the quality of the generated poem in human perspective. Specifically, we invite 57 participants who are native mandarin Chinese speakers to score the poems. During the evaluation, we randomly sample and show 15 poems written by different methods,

Table 3: The performance comparison of different methods on the COCO captions in terms of the BLEU scores and human evaluation scores.

| Method | BLEU-2 | BLEU-3 | BLEU-4 | Method | Human score |
|--------|--------|--------|--------|--------|-------------|
| MLE | 0.781 | 0.624 | 0.589 | SeqGAN | 3.44 |
| SeqGAN | 0.815 | 0.636 | 0.587 | RankGAN | 4.61 |
| RankGAN | **0.845** | **0.668** | **0.614** | Human-written | **6.42** |

Table 4: Example of the generated descriptions with different methods. Note that the language models are trained on COCO caption dataset without the images.

| Human-written |
|---|
| Two men happily working on a plastic computer. |
| The toilet in the bathroom is filled with a bunch of ice. |
| A bottle of wine near stacks of dishes and food. |
| A large airplane is taking off from a runway. |
| Little girl wearing blue clothing carrying purple bag sitting outside cafe. |
| SeqGAN (Baseline) |
| A baked mother cake sits on a street with a rear of it. |
| A tennis player who is in the ocean. |
| A highly many fried scissors sits next to the older. |
| A person that is sitting next to a desk. |
| Child jumped next to each other. |
| RankGAN (Ours) |
| Three people standing in front of some kind of boats. |
| A bedroom has silver photograph desk. |
| The bears standing in front of a palm state park. |
| This bathroom has brown bench. |
| Three bus in a road in front of a ramp. |

including RankGAN, SeqGAN, and written by human. Then, we ask the subjects to evaluate the quality of the poem by grading the poem from 1 to 10 points. It can be seen in Table 2, human-written poems receive the highest score comparing to the machine-written one. RankGAN outperforms the compared method in terms of the human evaluation score. The results suggest that the ranking score is informative for the generator to create human-like sentences.

### 4.3   Results on COCO image captions

We further evaluate our method on the large-scale dataset for the purpose of testing the stability of our model. We test our method on the image captions provided by the COCO dataset [19]. The captions are the narrative sentences written by human, and each sentence is at least 8 words and at most 20 words. We randomly select $80,000$ captions as the training set, and select $5,000$ captions to form the validation set. We replace the words appeared less than 5 times with UNK character. Since the proposed RankGAN focuses on unconditional GANs that do not consider any prior knowledge as input, we train our model on the captions of the training set without conditioning on specific images.

In the experiment, we evaluate the performance of the language generator by averaging BLEU scores to measure the similarity between the generated sentences and the human-written sentences in the validation set. Table 3 shows the performance comparison of different methods. RankGAN achieves better performance than the other methods in terms of different BLEU scores. Some of the samples written by humans, and synthesized by the SeqGAN and the proposed model RankGAN are shown in Table 4. These examples show that our model is able to generate fluent, novel sentences that are

Table 5: The performance comparison of different methods on Shakespeare's play *Romeo and Juliet* in terms of the BLEU scores.

| Method | BLEU-2 | BLEU-3 | BLEU-4 |
|--------|--------|--------|--------|
| MLE | 0.796 | 0.695 | 0.635 |
| SeqGAN | 0.887 | 0.842 | 0.815 |
| RankGAN | **0.914** | **0.878** | **0.856** |

not existing in the training set. The results show that RankGAN is able to learn effective language generator in a large corpus.

We also conduct human study to evaluate the quality of the generated sentences. We invite 28 participants who are native or proficient English speakers to grade the sentences. Similar to the setting in previous section, we randomly sample and show 15 sentences written by different methods, and ask the subjects to grade from 1 to 10 points. Table 3 shows the human evaluation scores. As can be seen, the human-written sentences get the highest score comparing to the language models. Among the GANs approaches, RankGAN receives better score than SeqGAN, which is consistent to the finding in the Chinese poem composition. The results demonstrate that the proposed learning objective is capable to increase the diversity of the wording making it realistic toward human-like language description.

### 4.4 Results on Shakespeare's plays

Finally, we investigate the possibility of learning Shakespeare's lexical dependency, and make use of the rare phrases. In this experiment, we train our model on the Romeo and Juliet play [28] to further validate the proposed method. The script is splited into $2,500$ training sentences and $565$ test sentences. To learn the rare words in the script, we adjust the threshold of UNK from 5 to 2. Table 5 shows the performance comparison of the proposed RankGAN and the other methods including MLE and SeqGAN. As can be seen, the proposed method achieves consistently higher BLEU score than the other methods in terms of the different n-grams criteria. The results indicate the proposed RankGAN is able to capture the transition pattern among the words, even if the training sentences are novel, delicate and complicated.

## 5 Conclusion

We presented a new generative adversarial network, RankGAN, for generating high-quality natural language descriptions. Instead of training the discriminator to assign absolute binary predicate to real or synthesized data samples, we propose using a ranker to rank the human-written sentences higher than the machine-written sentences relatively. We then train the generator to synthesize natural language sentences that can be ranked higher than the human-written one. By relaxing the binary-classification restriction and conceiving a relative space with rich information for the discriminator in the adversarial learning framework, the proposed learning objective is favourable for synthesizing natural language sentences in high quality. Experimental results on multiple public datasets demonstrate that our method achieves significantly better performance than previous state-of-the-art language generators. In the future, we plan to explore and extend our model in many other tasks, such as image synthesis and conditional GAN for image captioning.

## Acknowledgement

We would like to thank the reviewers for their constructive comments. We thank NVIDIA Corporation for the donation of the GPU used for this research. We also thank Tianyi Zhou and Pengchuan Zhang for their helpful discussions.

## Footnotes

[2]The synthetic data and the oracle model (LSTM model) are publicly available at https://github.com/LantaoYu/SeqGAN

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
