[Reviews · NeurIPS 2017]

Reviewer 1



For generative adversarial network (GAN) training, the paper presents an approach for replacing the binary classifier in the discriminator with a ranking based discriminator. This allows better training especially for problems which are not binary, such as (discrete) sequence generation, as e.g. in sentence generation. Strength: - Novel, well motivated approach - Extensive experiments on four datasets, a synthetic dataset, Chinese poem generation, coco image caption generation, and Shakespeare’s plays generation. - The results show improvements over the recent RankGAN [33], as well as a maximum likelihood estimation (MLE) baseline. The paper shows the improvements w.r.t. automatic as well as human evaluation measures. - The paper is clear and to my understanding sound. Weaknesses: 1. Related Work: As the available space allows it, the paper would benefit from a more detailed discussion of related work, by not only describing the related works, but also discussing the differences to the presented work. 2. Qualitative results: To underline the success of the work, the paper should include some qualitative examples, comparing its generated sentences to the ones by related work. 3. Experimental setup: For the Coco image cations, the paper does not rely on the official training/validation/test split used in the COCO captioning challenge. 3.1. Why do the authors not use the entire training set? 3.2. It would be important for the automatic evaluation to report results using the evaluation sever and report numbers on the blind test set (for the human eval it is fine to use the validation set). Conclusion: I hope the authors will include the coco caption evaluation server results in the rebuttal and final version as well as several qualitative results. Given the novelty of the approach and strong experiments without major flaws I recommend accepting the paper. It would be interesting if the authors would comment on which problems and how their approach can be applied to non-sequence problems.

Reviewer 2



The paper proposes an alternative parameterization of the GAN discriminator based on relative ranks among a comparison set. The method shows better quantitative results for language generation on three (relatively small) language datasets. Mathematically, there are some potential problems of the proposed method. - Firstly, the original GAN identified the equilibrium of Eqn. (1) only for R(s|U,C) being a proper probability over a Bernoulli variable. However, the definition in Eqn. (4) does not guarantee this condition. Thus, it is difficult to see whether the proposed criterion can ensure the generator distribution to match the true data distribution. - However, if one regards the expected log probability in Eqn. (4) as a definition of energy, the practically used criterion Eqn. (7) is more similar to the energy-based GAN. From this perspective, if one completes the two expectations on U and C omitted in Eqn. (7), (i.e., the full formula for the a term should be E_{s ~ P_h} [ E_{U ~ bar{U}} E_{C ~ bar{C}} log R(s|U,C) ] where bar{U} and bar{C} represent all possible reference and comparison sets respectively), and plugs Eqn. (4) into the full formula, it becomes E_{s ~ P_h} [ E_{U ~ bar{U}} E_{C ~ bar{C}} ( E_{u ~ U} log P(s|u,C) ) ] <= E_{s ~ P_h} [ log P(s) ], which is a lower bound of the ranker-defined likelihood. In other words, the energy function is defined as a lower bound of the likelihood, which is a little bit strange and not conventional. - Putting things together, I don’t really understand what the proposed criterion is doing, and why it leads to better empirical performances. In terms of the experiment result, despite the good quantitative results, it would be nice if some qualitative analysis were conducted to understand why and how the generated sentences could achieve higher scores. Also, is the result sensitive to the size of the comparison set |C| or the size of the reference set |U|? In summary, the proposed method achieves nice empirical results in improving language generation under the GAN setting. However, given the content of the paper, it is still not clear why this method should lead to such improvement.

Reviewer 3



This paper propose to use ranking loss to replace original binary classification loss in GAN, which is novel. All the evaluation scores are better than state-of-art sequence generating GAN, however, there is no sample of generated text, which is hard to say whether this method is overfitting the metric.